# Data Link with a High-Power Pulsed Quantum Cascade Laser Operating at the Wavelength of 4.5 µm

**DOI:** 10.3390/s21093231

**Published:** 2021-05-07

**Authors:** Janusz Mikołajczyk

**Affiliations:** Institute of Optoelectronics, Military University of Technology, Kaliskiego 2, 00908 Warsaw, Poland; janusz.mikolajczyk@wat.edu.pl

**Keywords:** free space optics, quantum cascade lasers, optical wireless communications, infrared detectors

## Abstract

This article is a short study of the application of high-power quantum cascade lasers and photodetectors in medium-infrared optical wireless communications (OWC). The link range is mainly determined by the transmitted beam parameters and the performance of the light sensor. The light power and the photodetector noise directly determine the signal-to-noise power ratio. This ratio could be maximized in the case of minimizing the radiation losses caused by atmospheric attenuation. It can be obtained by applying both radiation sources and sensors operated in the medium infrared range decreasing the effects of absorption, scattering or scintillation, beam spreading, and beam wandering. The development of a new class of laser sources based on quantum cascade structures becomes a prospective alternative. Regarding the literature, there are descriptions of some preliminary research applying these lasers in data transmission. To provide a high data transfer rate, continuous wave (cw) lasers are commonly used. However, they are characterized by low power (a few tens of mWatts) limiting their link range. Also, only a few high-power pulsed lasers (a few hundreds of mWatts) were tested. Due to their limited pulse duty cycle, the obtained modulation bandwidth was lower than 1 MHz. The main goal of this study is to experimentally determine the capabilities of the currently developed state-of-the-art high-power pulsed quantum cascade (QC) lasers and photodetectors in OWC systems. Finally, the data link range using optical pulses of a QC laser of ~2 W, operated at the wavelength of ~4.5 µm, is discussed.

## 1. Introduction

The mid-infrared wavelength range (MWIR) is very promising for many applications. The main ones are chemical sensing and spectroscopy developed for the pharmaceutical sector, biomedical analysis, atmospheric chemistry, detection of dangerous substances and materials, etc. Regarding these configurations, there is a need to use radiation sources characterized by a continuous wave (cw) and room-temperature (RT) operation, high spectral power density, single-mode emission, and high integration. There are also numerous other applications of MWIR radiation in defense, security, and in the “last mile” optical wireless communications. Concerning these technologies, the light source should have a compact size, high efficiency, high output power, high reliability, and design flexibility to a specific infrared range. Considering the mentioned requirements, quantum cascade lasers are ideal sources of mid-infrared radiation. Nowadays, some scientific works have been performed to design an OWC system using QC lasers.

Application of MWIR quantum cascade lasers in the OWC systems allows operating in the “transparent window” of the atmosphere (low absorption). Comparing commercially available systems operated at the near-infrared (NIR) or the short-infrared wavelength (SWIR) range reveals that the longer wavelength also decreases the attenuating influence of other optical phenomena, for example, beam scattering, beam wandering, loss of spatial coherence, or scintillation. Most often the goal of the OWC system design is to ensure a high data link capacity. To this end, cw lasers are used to apply non-limited coding formats of light signals. Due to a low carrier lifetime of a QC laser upper-state (in the range of a few ps), the bandwidth above 100 GHz is theoretically possible. However, it requires radio frequency (RF) modulation of the QC laser radiation. Nowadays, RF optical signals are obtained by driving a QC laser with a bias-T circuit. Several works presented a few-GHz direct modulation [1]. Faster modulation speeds also have been reported, but this requires a specially constructed lasing structure to inject the RF signal, or the use of cryogenic cooling [2]. This speed can be improved by designing the device housing and optimizing the laser cavity [3]. Therefore, the main goal of the performed research is to design QC lasers with direct GHz modulation. Regarding the description of the obtained results, there are no comments about power modulation parameters such as power efficiency and depth, critical for communications applications. The conducted literature reviews also show that there is no information about the application of a high-power pulsed QC laser in an OWC system [4]. It results mainly from a low duty cycle (dc) of its generated light pulses limiting the format of signal coding. This low dc is due to the lasing structure heating with high driving currents.

Regarding the case of the OWC receiver, there are many radiation sensors that can be used to register data signals. The selection of these sensors is based on their detectivity, dynamic response, time constant, operation temperature, reliability, cost, etc. The technology evaluation of infrared detectors was described in many publications [5,6,7,8]. 

Nowadays, some technologies based on mercury cadmium–telluride (MCT), quantum well infrared photoconductors (QWIP), and strained layer superlattices (SLS)–have been tested for data transmission in free space. Some properties of these technologies are listed in Table 1.

Mercury cadmium–telluride (MCT or HgCdTe) is the most often used detector technology tested in an OWC system operated in the MWIR range. MCT detectors are highly advanced and ensure more than an order of magnitude higher detectivity in comparison to the value predicted by Rule 07 [16]. They have a high quantum efficiency (QE) due to the high absorption coefficient for the photon energy corresponding to the fundamental energy gap. The QE is the best one to compare other technologies. However, MCT detectors have large dark currents. There are also some production problems due to a weak mercury–telluride bond. This causes poor material uniformity, surface instability, a low yield, and high-cost processing.

A quantum well infrared photoconductor (QWIP) is an alternative detector characterized by better thermal stability and higher uniformity. What is more, it provides lower dark currents, higher detectivities, and easier fabrication. The short carrier lifetime of a QWIP ensures achieving a faster response (>100 GHz bandwidth). However, the QE is limited by no optical transition for the normal incidence of light. To minimize this effect, a frontside scattering filter (reflecting gratings) is applied but some light intensity is still guided by the residual substrate. QWIP detectors should work at a low temperature, increasing the QE and decreasing the dark current. The IR response speed and low power consumption are interesting for the construction of commercial OWC receivers. 

The SLS technology provides a responsivity comparable to MCT, but it is in an early stage of development. The applied developed techniques of these structures ensure precise control of the layer thicknesses and interfaces. Due to a low dark current, normal incidence absorption, room temperature operation, and band-gap, tunable type-II superlattice (T2SL) detectors are desired in an OWC system. However, a fairly narrow effective band gap between the minibands makes this technology less effective for shorter wavelength applications. Despite this, SLSs can be a future-proof alternative to MCT as the third generation of infrared detectors. 

A detailed analysis of the infrared detector technologies, which are currently used in OWC systems and also are being developed, is presented by Downs et al. [7] and Chen et al. [8]. Additionally, quantum dot infrared photodetectors (QDIPs), quantum cascade detectors (QCDs), and interband cascade infrared photodetectors (ICIPs) are described also.

Regarding Table 2, the parameters of three defined groups of QC lasers and photodetectors used to transmit optical signals are listed. Group ‘A’ presents some lasers tested for transmission of analog or digital signals. They have a special lasing structure (with RF signal injection) driven by the bias-T circuit. These features provide for the lowest driving currents and the highest bandwidth. The lasers listed in group ‘B’ are ultra high-power ones with phased arrays (QCL–PA), photonic crystal distributed feedback (QCL–PCDF), and master-oscillator power-amplifier (QCL–MOPA) configurations. Their power level is a few tens of Watts, driving the current up to 25 A. Their preliminary test showed tens of kHz modulation frequencies.

Group ‘C’ describes the results of the author’s work to design the data link up to 2 km with the transfer of 10 Mb/s for real operation scenarios. Regarding that setup, QC lasers operated at a wavelength of 9.35 µm with a peak power of 300 mW, and a maximal modulation speed of 4 MHz. 

Additionally, some high-power QC lasers are available commercially. Pranalytica, for example, offers a QC laser operated at the wavelength of 4.6 μm with an average power above 4 W [25]. It can be modulated with a standard laser driver up to 100 kHz or with a special switching unit up to 1 GHz [26]. However, there is no further information about this unit. 

The main goal of this study is to experimentally determine the capabilities of currently developed state-of-the-art high-power pulsed QC lasers. The described technology enables designing data links with high-power lasing structures operating at the wavelength of 4.5 µm with a peak pulse power of 3 W. Concerning the analytical part of this work, the performance parameters of the designed OWC system for different weather conditions are analyzed. Finally, some laboratory tests of this link are presented. The obtained results are unique considering both power and modulation speeds. Regarding frequencies of 5 MHz and 7 MHz, the optical peak power is 1.27 W and 0.64 W, respectively.

## 2. Materials and Methods

### 2.1. Analysis of the OWC System

The modulated light is emitted by a transmitter and registered by a receiver (Figure 1). To determine the data link range, the signal into the noise power ratio (determined by the light power registered by the receiver and ‘floor’ signal generated for no-light conditions) is analyzed. The registered power $P_{R}$ is calculated by [27]:(1)$$P_{R}=P_{L}\frac{D_{d}^{2}}{{\left(D_{0}+\theta_{div}L\right)}^{2}}e^{-\gamma \left(\lambda \right)L},$$
where $P_{L}$ is the laser optical power, γ(λ) is the extinction coefficient, L is the link distance, $\theta_{div}$ is the beam divergence, $D_{0}$ and $D_{d}$ are the aperture diameters of the transmitter and the receiver. The extinction coefficient usually determines a light attenuation caused by both absorption and scattering effects.

The absorption describes the light interaction with air gases. There are some “spectral windows” in which this effect is minimal (Figure 2). 

Analyzing the scattering effects, the radiation losses can be estimated using visibility (*V*is) and different models [28,29]. The visibility is combined with well-defined weather conditions described by the International Visibility Code [17]. 

Regarding Figure 3, the comparison of the extinction coefficient for two wavelengths (SWIR—1.55 µm and MWIR—4.5 µm) and two selected weather seasons is presented. The wavelength of 1.55 µm is used as a reference in the commercially available OWC systems. The coefficients were determined using PcModwin 6.0 software (Ontar Corporation, North Andover, MA, USA) for the urban area and two seasons. Weather conditions were defined by temperatures of 20.5 °C and −1.5 °C, and water vapors of 13.4 g/m^3^ and 3.4 g/m^3^ (RH~70%) for summer and winter, respectively. 

It can be seen that the scattering is decreasing with an increase in visibility. However, the radiation losses can be determined by its absorption for longer wavelengths (MWIR) and by its scattering for shorter ones (SWIR). 

More complicated analyses should be performed in the case of attenuation by radiation or advection fogs. To compare, advection fog is characterized by a linear attenuation wavelength dependence and the radiation fog by a quadratic one [30]. Additionally, the fog increases the air relative humidity (RH) [31]. The attenuation coefficient of radiation at two wavelengths is listed in Table 3.

Further scattering attenuation is caused by the presence of rain, snow, and dust. Except for these phenomena, the relation between the size of the compared particles and the analyzed radiation wavelength indicates no spectral benefits [32]. 

The optical link performance also is modified very often by air turbulence. This phenomenon is based on a random variation of the refractive index with fluctuations in temperature and pressure. Therefore, beam wander, beam spreading, or scintillation can be observed. During these conditions, the beam profile is redistributed. The change in temporal and spatial light direction, and fluctuations in its power and phase, are registered. The origin of turbulence and its optical effects are precisely analyzed in [33]. Therein is described a relation of the refractive index to the light wavelength, pressure, humidity, and temperature. The strength of turbulence, the refractive index structure constant—C_n_^2^, also was defined. The C_n_^2^ values range from about 10^–12^ to 10^–16^ m^–2/3^. Values above 10^–13^ m^–2/3^ most often determine high turbulence in the atmosphere. Regarding [34], the influence of the atmosphere on laser beam propagation for three turbulence effects was analyzed. Concerning the case of beam wander, the generated optical signal fluctuations have a frequency of a few kHz and can be minimized using a tracking device. Regarding short-term beam spreading, this effect is typically neglected. The wavelength dependency of these turbulences is not strong. Actually, OWC systems are most often affected by scintillation. Random wave front interference causes peaks and dips in the high-dynamic receiver output signal. This effect can be minimized using, for example, aperture averaging, diversity techniques, adaptive optics, or some modulation techniques [35]. It also was shown that an operation at longer wavelengths allows for better transmission during turbulence [36]. However, strong turbulence causes comparable radiation attenuation for NIR, SWIR, and MWIR ranges. 

The OWC system performances also can be limited by a pointing error specifying the misalignment between a transmitter and a receiver. Thermal expansions, strong wind, and weak earthquakes are the main factors of building sway and mechanical vibration moving the OWC transceivers from the common line of sight [37]. The average received signal is decreased and, consequently, the bit-error-rate (BER) is increased.

Analyzing the SNR level, the main source of noise power is the receiver noise. It is generated in a photodetector and its front-end electronics. The photodetector noise is usually described by the Noise Equivalent Power (NEP): (2)$$NEP=\frac{\sqrt{A_{d}\Delta f}}{D^{*}},$$
where $D^{*}$—is the detectivity, $A_{d}$ is the active surface of the photodetector, and Δf is the noise bandwidth. The NEP value determines the fundamental detection limit of the transmitter signal. This limit, however, is increased by front-end noises. The total power noise $N_{T}$ is equal to:(3)$$N_{T}=N_{det}G+N_{f-e},$$
where $N_{det}$ is the photodetector noise signal, G is the signal gain of the photodetector front-end, and $N_{f-e}$ is the front-end noise contribution.

### 2.2. Link Budget

The link budget was made for an OWC system equipped with a pulsed QC laser (numbered as #356) developed at the Institute of Electron Technology (Warsaw, Poland). The laser is characterized by a wavelength of ~4.5 µm and high optical power (peak power above 3 W) (Figure 4). It can work at a maximum duty cycle of 8%.

The laser beam was formed using an off-axis parabolic mirror with a 3-inch diameter and a 2-inch focal length, obtaining a divergence of less than 1 mrad. This setup allows collecting ~75% of the laser power. The laser pulses were focused on the MCT photodetector using a 4-inch parabolic mirror. The photodetector was mounted with the designed front-end electronics in a detection module manufactured by the VIGO System company [38]. 

The applied photodiode is composed of MCT heterostructures as a monolithic chip with integrated optical, detection, and electronic functions providing a concentration of radiation, enhanced absorption, efficient and fast photon collection, suppression of thermal generation, and low parasitic impedances. 

The photodiode heterostructure is integrated with an immersion lens as an effective optical concentrator. Regarding the hemispherical immersion lens, the optical area is increased *n*^2^-times more than the physical one, where *n* is the refractive index of the lens material. This solution allows for the decreasing of an absorber volume and, thus, a reduction in the thermal generation and recombination of carriers (lower noise power). To provide the required operating temperature of the photodiode, below 200 K, the detector is placed on four-stage Peltier coolers. The cooler is controlled with a special driver to stabilize the detector operating temperature at variable ambient ones up to ~300 K. The current signal from the photodiode is received by a broadband two-stage amplifier (up to 1 GHz) with a gain of 10 kV/A (Figure 5). 

The electronic circuitry provides optimized conditions for the photodiode operation, i.e., the constant voltage bias and current mode readout [40]. The first stage is a transimpedance amplifier of a low input resistance based on an operational amplifier OPA 847, characterized by a low input noise voltage of 0.85 nV/Hz^1/2^ and a moderate input noise current of 2.5 pA/Hz^1/2^. The second stage is a ~20 dB voltage amplifier with 50 Ω output resistance. The parameters of the applied photodiode are listed in Table 4. 

The analyses of the data link range were performed using an analytical model based on dependencies described in *The Infrared Handbook* (by W. L. Wolfe, G. J. Zissis) and *Field Guide to Atmospheric Optics* (L. C. Andrews) [41,42]. Regarding this model, the input data of the parameters of the OWC system components and atmosphere attenuation were imported from Figure 2 and Table 4. No influence of pointing errors was assumed. The registered radiation power was determined, taking into consideration, e.g., laser pulse power, beam geometric losses, the reflectivity of mirrors, and transmission of the atmosphere (the extinction coefficient and the refractive index structure constant).

The noise level is defined by the noise of the optical receiver and the signal fluctuations due to turbulence. 

Considering Figure 6, some obtained characteristics of the SNR for different scattering effects (extinction coefficient) and scintillation (no-scintillation: C_n_^2^ = 10^−17^ m^−2/3^ and strong: C_n_^2^ = 10^−14^ m^−2/3^) were presented. Regarding the analyzed wavelengths, there was no difference in the link range (blue line) at this scintillation level. Scintillation was ideally neglected to highlight the impact of scattering for different weather conditions (*clear*: *V*is > 6 km, γ = 0.25 km^−1^; *light haze*: *V*is > 4 km, γ = 0.75 km^−1^; *haze*: *V*is > 2 km, γ = 1.25 km^−1^). These results can be recalculated to obtain the BER for different coding formats. It was shown that the described link theoretically ensures transmission of data at the distance of ~5 km with a SNR = 10 with no-scintillation and a light haze.

## 3. Results 

During tests, the lasing structure was placed in the housing LLH model (Alpes Lasers SA). The triggering signals for a laser driver were generated using a pattern signal Picosecond 12,000 model generator. To monitor laser current pulses, the current probe model CT-1 from Tektronix was applied. The registered pulses were processed using an MSO 64 model oscilloscope (Tektronix) with a built-in ‘eye diagram’ toolbox. The testing setup is presented in Figure 7.

### 3.1. QC Laser Research for OWC Application

The main goal of the preliminary tests was to define the power-time limitations of the emitted light pulses. Increasing their duration caused a transient decrease in the laser pulse amplitude (Figure 8a). This effect is determined by the laser structure heating during supplying the current. The shortest light pulse (~5 ns) was obtained for an 18 ns triggering signal. However, its amplitude was limited by the pulse rise time. The shapes of the driving current and the light pulses were coincidental but some differences in the rising pulse part were observed. The light pulse had a slower rise time and started later (delay time of ~14 ns) (Figure 8b). It is due to the time needed to exceed the lasing threshold current to generate population inversion in the laser structure. This delay also adversely influences the modulation rate and jitter. There also are observed amplitude oscillations generated by the impedance mismatching of the current driver and the lasing structure, which is noticeable itself especially for high currents. Practically, these current oscillations generate heat and can damage the laser structure. 

Figure 9 presents some registered pulses for different driving currents and pulse frequencies. The strong influence of both parameters on the shape of emitted pulses is observed. The current decreasing and frequency increasing led to a decrease in the laser pulse amplitude. Additionally, shortening of the optical pulses also was noticed. Regarding 5 MHz and 7 MHz pulse frequencies, the peak pulse level was decreased by about 30% (to the level of 1.27 W) and about 65% (to the level of 0.64 W), respectively.

Regarding data transmission, laser power stability is also important. This stability was tested for different pulse parameters (power, frequency, time duration). Considering Figure 10, the deviation range in the optical pulse amplitude for a 1.71 A driving current and for three frequencies (20 kHz, 3 MHz, and 5 MHz) are presented. Although for higher frequencies the light pulse level was decreasing, the power stability was improving.

Preliminary generation of on-off keying signals was performed by sending a frame containing several pulses (BURST mode). The BURST consisted of 16 pulses (time duration of 18 ns, period time of 100 ns) with a repetition rate up to the frequency of 2.5 MHz (Figure 11a). That range of repetition rate was limited by the efficiency of the cooling laser setup determined by Peltier module parameters mounted in the LLH housing. During the tests, the maximum current of this module (2 A) was reached. The minimum time period between two pulses was ~25 ns (three registered light pulses in the time period of 75 ns) (Figure 11b). The tested configuration allowed generating pulses with a time resolution of 25 ns, but the main limitation was the mentioned laser heating.

### 3.2. Transmitter with Pulsed QC Laser

The final step of the transmitter tests was to analyze emitted PRBS signals in RZ format coding. To this end, a 10-bits frame (1000010000_BIN_) was generated. Figure 12 presents registered eye diagrams for three modulation speeds (5 MHz, 6 MHz, and 7 MHz). 

An increase in the pulse frequency decreases the pulse amplitude and the eye width. These parameters correspond with the previously defined capabilities of the laser’s structure switching. Regarding the configured OWC system, the maximum modulation speed of 7 MHz was obtained. 

## 4. Conclusions

The paper experimentally determines the capabilities of the currently developed state-of-the-art high-power pulsed QC lasers and photodetectors in OWC systems. The interest in this technology results from the theoretical possibility of obtaining a higher link capacity (a high modulation rate of lasers) and a longer data range (a better atmospheric transparency). 

The performed analyses and tests of the constructed OWC system were provided to define its performance. Its estimated data range was 5 km, which was below the distance reported for some commercially available OWC systems operating in NIR or SWIR spectral ranges. However, in these systems, both a high data rate and long link range are usually ensured by applying a few lasers with low beam divergence in the transmitter and increasing the size of the receiver aperture. Regarding the designed system, the mentioned range was determined for a single QC laser without any advanced modulation techniques (only on–off keying), receiving optics with a 4-inch diameter, and a beam divergence of 1 mrad. Additionally, this result was estimated for the visibility of 4 km. 

The test results showed that for this type of laser, light pulses are shorter than the driving current ones. It results from both the rise time of the driving current pulses and the time needed to ensure population inversion in the lasing structure. This limits the modulation frequency of pulses. Some oscillations of the driving current coming from the no-resistance character of the QC laser were observed. Any inductance and capacitance also are very critical for ultra-high current signals and can damage the lasers. High pulse duty cycles cause a large amount of local heat generation in the laser structure by the high driving currents. It changes the operating point of the laser, usually increasing the threshold current and decreasing the light power. The modulation speed of the laser pulses also was limited. Despite the above-mentioned drawbacks, the described data link can be an effective communication device for many applications in which the data rate is less important than the link range. 

The modulation rate (up to 7 MHz) and peak power (above 0.6 W) of the light pulses are the first results obtained using a high-power QC laser operated at room temperature. They have defined new opportunities for long-range optical wireless communications. 

## Figures and Tables

**Figure 1 sensors-21-03231-f001:**
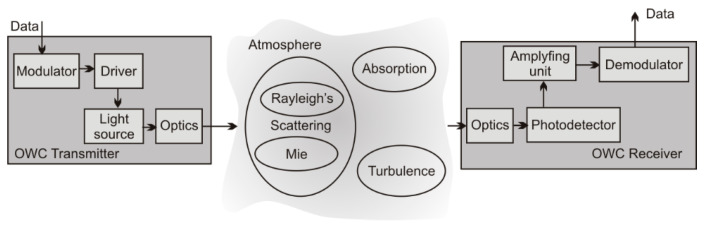
Block scheme of the OWC system.

**Figure 2 sensors-21-03231-f002:**
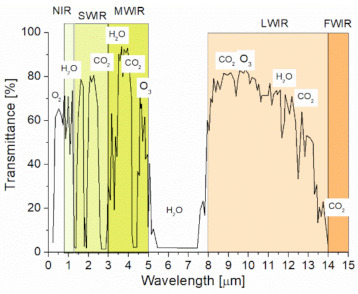
Spectral atmospheric transmittance.

**Figure 3 sensors-21-03231-f003:**
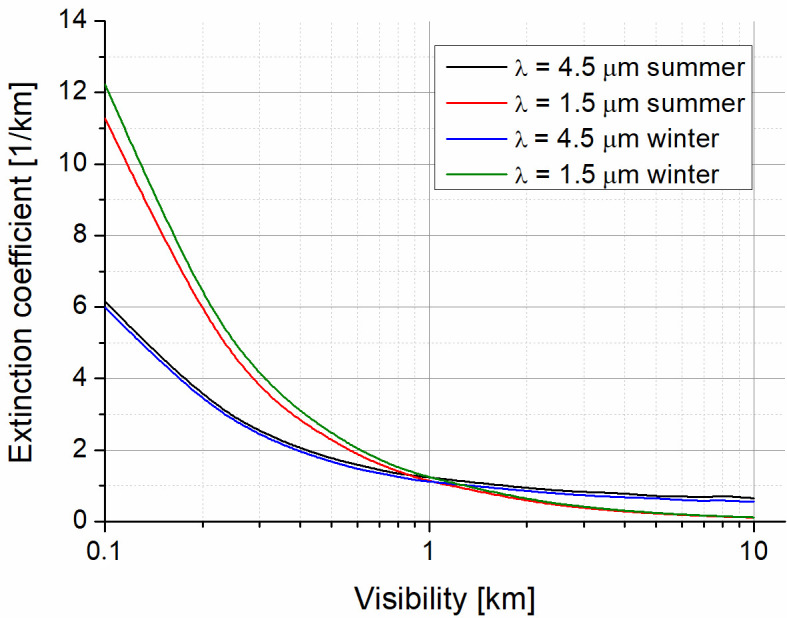
Extinction coefficient for two analyzed wavelengths.

**Figure 4 sensors-21-03231-f004:**
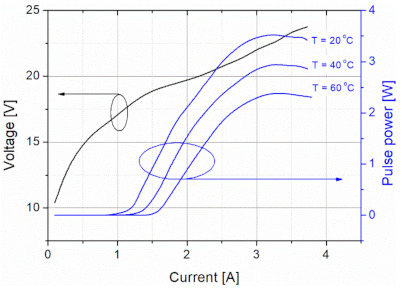
Characteristics of pulsed QC laser (#356).

**Figure 5 sensors-21-03231-f005:**
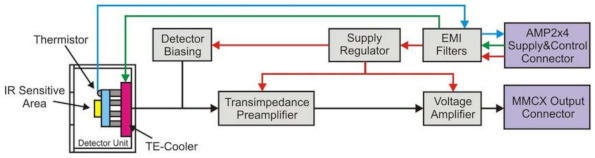
Schematic diagram of the detection module [39].

**Figure 6 sensors-21-03231-f006:**
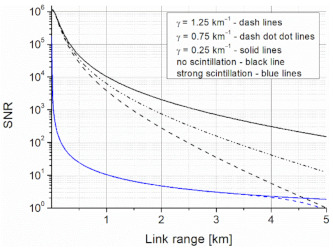
SNR values versus link range.

**Figure 7 sensors-21-03231-f007:**
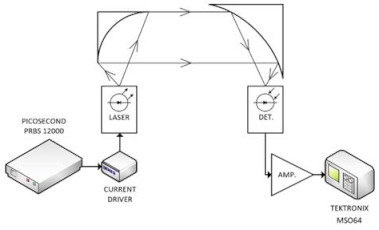
The block scheme of the OWC lab system with the testing setup (DET.—MCT photodetector, AMP.—detector amplifier).

**Figure 8 sensors-21-03231-f008:**
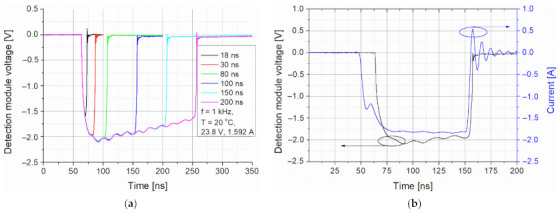
(**a**) Optical signals registered by the detection module for different pulse times and (**b**) shapes of driving current and light signals.

**Figure 9 sensors-21-03231-f009:**
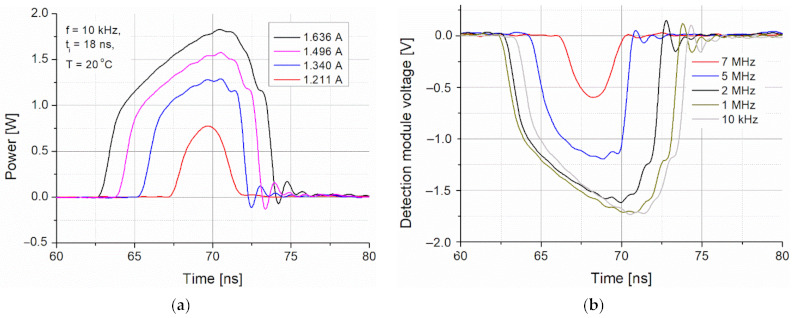
(**a**) Power of optical signals for different driving currents and (**b**) shapes of light pulses for different driving frequencies.

**Figure 10 sensors-21-03231-f010:**
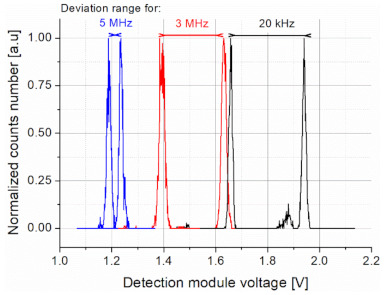
Deviation ranges of the optical pulse amplitude for different repetition rates.

**Figure 11 sensors-21-03231-f011:**
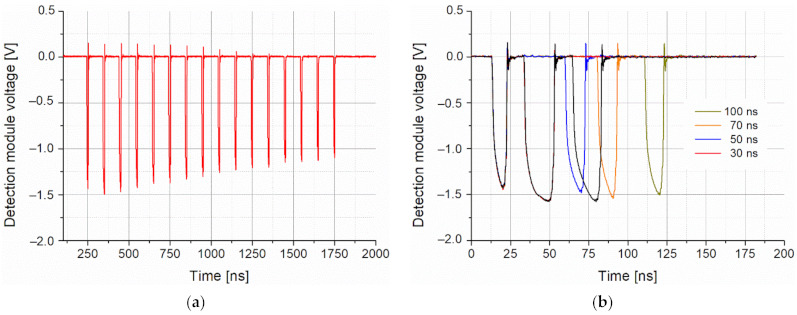
(**a**) Frame with 16 light pulses and (**b**) two registered pulses for different time periods.

**Figure 12 sensors-21-03231-f012:**
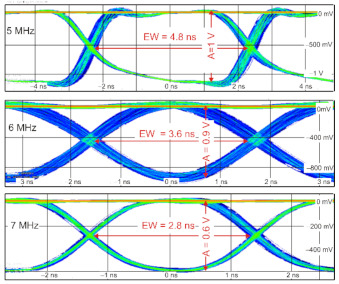
Eye diagrams for different modulation speeds: EW–eye width, A–“one” level amplitude.

**Table 1 sensors-21-03231-t001:** Essential properties of some infrared detectors.

	MCT	QWIP	SLS
Technological status	advanced	commercial	research
Quantum Efficiency	high	low	medium
Spectral range	wide	narrow	wide
Detectivity [cmHz^1/2^/W] (Temperature)	8.7 × 10^10^ (300 K) [9]	1 × 10^11^ (78 K) [10] 2 × 10^8^ (210 K) [11]	7.1 × 10^11^ (150 K) [12] 4.6 × 10^9^ (300 K) [13]
Response time	from tens/hundreds of ps up to a few ns [8]	a few ps [14]	a few ns [15]

**Table 2 sensors-21-03231-t002:** MWIR radiation sources and detectors applied to transmit optical signals in a free space.

Group	Wavelength [µm]	Modulation	Power/ Current	Mode Type	Sensor	Lit.
A	8	2.5 Gb/s	10 mW (0.3 + ?) A	Bias-T (cw OWC)	MCT	[17]
	9.7	Analog 1.5 GHz	10 mW 0.02 A	RF injection (cw)	MCT	[1]
	4.65	Analog 23 MHz PAM-4 4 GB/s	60 mW (0.18 + 0.35) A	Bias-T (cw OWC)	MCT	[18]
	4.7	Analog 30 MHz	12 mW (0.35 + 0.1) A	Bias-T (cw OWC)	MCT	[19]
	4.8	Analog 10 MHz	200 mW (0.65 + 0.4) A	Bias-T (cw OWC)	T2SL	[20]
	4.6	Analog 1 MHz	2.2 W 1.5 A	Pulsed (76%)	?	[21]
B	4.8	Analog 50 kHz	20 W	QCL-PA	MCT	[22]
	4.36	Analog 20 kHz	40 W	QCL-PCDF	InSb	[23]
	9.2–9.8	Analog 10 kHz	2 ÷ 10 W	QCL-MOPA	MCT	[24]
C	9.35	Analog 4 MHz	300 mW	Pulsed	MCT	[4]

**Table 3 sensors-21-03231-t003:** Attenuation coefficient by air relative humidity (HITRAN database).

Wavelength [µm]	Attenuation [1/km]		
	RH = 50%	RH = 70%	RH = 100%
1.5	0.014	0.02	0.028
4.5	1.23	1.31	1.42

**Table 4 sensors-21-03231-t004:** Parameters of the detection module.

Parameter	Value	Spectral Characteristics
Active area	1 × 1 mm^2^	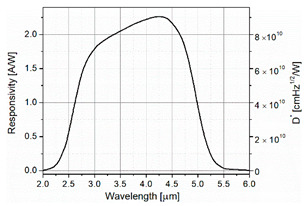
Detector resistance	60 kΩ	
Current noise density	2.4 pA/√Hz	
Responsivity	2.25 A/W	
Detectivity D*	9 × 10^10^ cm√Hz/W	

## Data Availability

Not applicable.

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
