# Peer review of "Data Link with a High-Power Pulsed Quantum Cascade Laser Operating at the Wavelength of 4.5 µm"

_sensors, 2021, doi:10.3390/s21093231_

Round 1
Reviewer 1 Report
See attached file.

Author Response
Dear Sir,
at first, I would like to thank you for your suggestions and opinion. They are very accurate and significantly increase the research level of my work. In the attached file, I sent my response.
Best Regards

Reviewer 2 Report
This is an interesting work focused on high power pulsed quantum cascade (QC) lasers’ development in OWC systems as a viable alternative to the establishment of more effective wireless links. Indeed, their operation at the emerging 4.5 μm, i.e. within the very promising Mid-infrared wavelength range (MWIR) makes this work potentially interesting, since this wavelength band is a prime candidate for the future high-speed wireless links. After the theoretical analysis performed along with the experimental results presented it has been demonstrated that the utilization of such lasers can be significantly efficient especially in terms of coverage area, which is a very critical issue for any modern wireless link. Additionally, the manuscript is well- organized and well-structured. However, for the sake of completeness I have the following remarks:
1) While major negative-side effects on outdoor OWC performance such as atmospheric turbulence, attenuation due to absorption and scattering, weather conditions have been reported in this work, there is no mention of pointing errors effect. Being another major limiting factor on outdoor OWC performance and availability the latter effect should be also mentioned. In this context, it should be also clarified that it has been ideally considered as a negligible one through the theoretical analysis performed.
2) Some more details about the origin and the consequences of atmospheric turbulence could be added.
3) In figure 3, please give if it is possible some more specific information (apart for summer/ winter) about parameter values or conditions that may affect these results (such as day/night, weather conditions, air temperature, urban or maritime environment, etc.)
4) Please explain in more detail how the distance at 4,5 km has been obtained. Note also that even in clear weather conditions turbulence-induced scintillation is unavoidable after 1km of propagation distance. Thus, for the sake of clarity I recommend that you clarify it for the case of black solid line (e.g. that scintillation effect for distances > 1km has been ideally neglected in order to highlight and focus on the impact of scattering).
5) Please check for typos (e.g. in line 206 it should be on-off keying (OOK) instead of on-of keying).
6) Some more discussions should be added in order to further highlight the contribution of this work along with the wireless applications that could harvest the benefits of the proposed laser technology.
To conclude, in my opinion the work is worth publishing especially after addressing the minor concerns described above.
Author Response

(The authors gave the same response as above.)

Reviewer 3 Report
Your work is interesting. However, I didn’t see the sensing application(s) strongly correlated to your system although you talked “Mid-Infrared Sensors and Applications”. Please strengthen this part, otherwise I recommend you to transfer your article to the other suitable journals.
For writing concerns, some grammars must be corrected and some expressions are not clear. If possible, you need to check the whole article again.
For instance:
- In Lines 15 and 31, the abbreviations of “cw” are applied. They will confuse the readers.
- In Line 55, “Its results” should be “It results”.
- In Lines 58 and 61, “(1¸6)” or “(7¸9)” is not a good expression.
4.. In Line 147, what is “MCT”?
- In Table 3, what is “9 1010” ?
For the technical concerns, please comment them:
- In Fig. 3, the scattering coefficient at wavelength of 1.5um is greater than that at 4.5um, especially at visibility < 1km. The illustration is not sufficient. Please comment it in detail. After the visibility > 1km, the phenomenon is changed. Please also comment it.
2.In Table 2, the attenuation [1/km] at wavelength of 1.5um is less than that at 4.5um. The lowest value is at RH=50%. Please comment both. Is this effect correlated to Fig. 3?
3. In Fig. 11, the eye diagrams for different modulation speed are listed. Please comment the performance of eye diagrams with speed. Please also add a reference or illustrate the extraction of eye diagram in eye height and eye width for the readers. As the speed is increased, the eye height and eye width are both decreased. What are the acceptable ranges or parameters in data link?
Author Response

(The authors gave the same response as above.)

Round 2
Reviewer 3 Report
Good revision!
Author Response
Dear Sir,
thank you very much for your help to make this paper better.
Best regards